# Far-Red-Light-Induced Morphology Changes, Phytohormone, and Transcriptome Reprogramming of Chinese Kale (*Brassica alboglabra* Bailey)

**DOI:** 10.3390/ijms24065563

**Published:** 2023-03-14

**Authors:** Yamin Li, Haozhao Jiang, Meifang Gao, Rui He, Xiaojuan Liu, Wei Su, Houcheng Liu

**Affiliations:** College of Horticulture, South China Agricultural University, Guangzhou 510642, China; yaminli@stu.scau.edu.cn (Y.L.);

**Keywords:** *Brassica alboglabra*, far-red light, morphology, phytohormone, RNA sequencing, mineral element

## Abstract

With far-red-light supplementation (3 W·m^−2^, and 6 W·m^−2^), the flower budding rate, plant height, internode length, plant display, and stem diameter of Chinese kale were largely elevated, as well as the leaf morphology such as leaf length, leaf width, petiole length, and leaf area. Consequently, the fresh weight and dry weight of the edible parts of Chinese kale were markedly increased. The photosynthetic traits were enhanced, and the mineral elements were accumulated. To further explore the mechanism that far-red light simultaneously promoted the vegetative growth and reproductive growth of Chinese kale, this study used RNA sequencing to gain a global perspective on the transcriptional regulation, combining it with an analysis of composition and content of phytohormones. A total of 1409 differentially expressed genes were identified, involved mainly in pathways related to photosynthesis, plant circadian rhythm, plant hormone biosynthesis, and signal transduction. The gibberellins GA_9_, GA_19_, and GA_20_ and the auxin ME-IAA were strongly accumulated under far-red light. However, the contents of the gibberellins GA_4_ and GA_24_, the cytokinins IP and cZ, and the jasmonate JA were significantly reduced by far-red light. The results indicated that the supplementary far-red light can be a useful tool to regulate the vegetative architecture, elevate the density of cultivation, enhance the photosynthesis, increase the mineral accumulation, accelerate the growth, and obtain a significantly higher yield of Chinese kale.

## 1. Introduction

Plants perceive ambient light signals via a series of photoreceptors to adjust their growth and development. Far-red light (FR, beyond 700 nm), once regarded as non-photosynthetically active radiation, is perceived by phytochromes (PHYs, PHYA-E). The FR or R:FR ratio is commonly known to induce shade tolerance or even shade avoidance syndromes. The typical responses include promoted apical dominance, increased leaf area but decreased leaf blade area, narrowed leaf angle, bud outgrowth, branching, and early flowering [1]. Generally, in open-field cultivation, these syndromes could lead to a severe reduction in crop yields due to the weakened plant robustness to pests and pathogens and the unevenly allocated resources on reproduction [2]. However, FR is now attracting research attention owing to its pivotal roles in photosynthesis [3], important features of modifying plant morphology [4], inducing transcriptional and metabolic reprogramming [5], and promoting yield of crops in protected horticulture [6].

FR induces profound changes in plant morphology. Higher doses of end-of-day FR (EOD-FR, 1.9 vs. 1.0 mmol·m^−2^) significantly increased the hypocotyl length of tomato rootstock seedlings [7]. EOD-FR led to obviously higher internode length of poinsettia plants [8]. A lower R:FR ratio (0.25 vs. 1.2) resulted in longer petioles of *Arabidopsis* [9]. The activity of XYLOGLUCAN ENDOTRANSGLUCOSYLASE/HYDROLASES (XTHs) could be enhanced by reducing the R:FR ratio which led to longer petioles [10]. Furthermore, FR triggers the elongation growth by PHYB inactivation which increases the activity and the abundance of PHYTOCHROME-INTERACTING FACTORS (PIFs) [11]. Subsequently, the PIFs promote the biosynthesis of auxin and the expression of growth-related genes which eventually results in cell elongation [12]. Meanwhile, FR was reported to induce expression of the gibberellin (GA) biosynthesis genes *GA3ox* and *GA20ox* [13] and a lower R:FR ratio (0.87 vs. 1.42 vs. 4.73) induced higher levels of GAs (GA_1_, GA_8_, GA_19_, GA_20_) and auxin (IAA), which contributed to longer internode length of sunflowers [14]. Besides, FR strongly influenced the leaf hyponasty and phototropism. EOD-FR obviously decreased the leaf angle of tomato plants [15]. Supplemental FR stimulated the upward leaf movement of tomato plants [16] and a lower R:FR ratio (0.2 vs. 4.2) induced hyponasty of *Arabidopsis* leaves [17]. The FR-induced plant morphological plasticity allows plants to capture light effectively.

Another important phenotypic response to FR is accelerated flowering. In many long-day plants such as *Arabidopsis thaliana*, *Campanula carpatica*, and *Gypsophila paniculata*, FR simultaneously increases the PHYA activity and decreases the PHYB/D/E activity, which directly or indirectly stabilizes the CONSTANS (CO) protein against degradation and consequently activates the transcription of *FLOWERING LOCUS T* (*FT*) and initiates plant flowing [18]. Both additional and substitutional FR shortened the days to flowering in snapdragon, while they had no effect on the flowering time of geranium, petunia, and impatiens [19]. Therefore, the induction of flower via FR seems to be species-dependent. 

Supplementary FR obviously increased shoot fresh weight and shoot dry weight of Chinese kale baby leaves after 6 days of treatment [20]. EOD-FR (30 min) markedly increased the plant fresh weight, shoot fresh weight, plant dry weight, and shoot dry weight of red-leaf and green-leaf lettuce [21], whereas day-FR (10 h) had no effects on lettuce biomass [22]. The biomass is commonly known to directly depend on photosynthesis. FR photons were found synergistically increase the photochemical efficiency of lettuce leaves with blue and red photons [23]. The canopy photosynthesis increased by substitutional FR (40%) was equal to the photosynthetically active radiation (PAR, 400–700 nm) [24]. Nevertheless, additional FR reduced the relative specific chlorophyll content (SPAD) [25] and decreased the contents of chlorophylls and carotenoids of lettuce [26,27]. Therefore, FR seems to have discrepant effects on the photosynthesis and the contents of photosynthetic pigments. 

Mineral nutrient is another aspect that strongly influences crop biomass and yield. The uptake and utilization of mineral elements could be dynamically altered by light signals. FR, red, and blue irradiation activate PHYA, PHYB, and CRYPTOCHROME 1 (CRY1) which induces ELONGATED HYPOCOTYL 5 (HY5) activity by suppressing activity of SUPPRESSOR OF PHYTOCHROME A1–CONSTITUTIVELY PHOTOMORPHOGENIC1 (COP1)–SUPPRESSOR (SPA1–COP1) complex. The HY5, a key component of light signaling that is responsible for the regulation of fundamental developmental processes of the plant cell [28], directly regulates nitrogen (N) uptake and assimilation though promoting expression of *NITRATE TRANSPORTER 2.1* (*NRT2.1*) and *NITRATE REDUCTASE 2* (*NIA2*), mediates sulfur (S) assimilation by inducing *ADENOSINE 5′-PHOSPHOSULPHATE REDUCTASE* 1/2 (*APR1/2*) expression, modifies copper (Cu) utilization via the modulation of *microRNA408* (*miR408*) expression [29], and influences potassium (K) uptake and accumulation by regulating the expression of *POTASSIUM TRANSPORTER 6/10/15/19* (*HAK6/10/15/19*) [30]. 

This study applied different dosages of FR in a plant factory, aiming to manipulate the growth, morphology, and mineral nutrient absorption of Chinese kale (*Brassica alboglabra* Bailey) and explore the underlying molecular mechanisms by co-analyzing transcriptome data and agronomic traits, which might further increase its yield and economic value.

## 2. Results

### 2.1. Plant Biomass and Morphological Indices Indicate an Elevated Growth of Chinese Kale under Supplementary Far-Red Light

In general, plants grown in higher far-red light condition (shade) shift to the reproductive growth earlier, at the expense of robustness. However, in the early stage of the experiment, we observed that the Chinese kale under far-red light had a relatively short growth period, with no decreased biomass compared to those grown under control light. To analyze the effects of far-red light on the Chinese kale growth, the plant biomass and morphological indices were measured in this study (Figure 1 and Table 1). 

At 45 days of light treatments, the budding rate of CK (control group) was only 11.1% (Figure 1); while that of FR-3 and FR-6 reached 30.6% and 45.8%, respectively, indicating an accelerated flower formation. Meanwhile, plant morphology of Chinese kale was observably changed by supplementary far-red light. The highest plant height and internode length were found in FR-6, with an increase of 123.3% and 135.3%, respectively. Those in FR-3 were 63.0% and 67.7% higher than in CK, respectively. Compared to CK, the plant displays in FR-3 and FR-6 were significantly increased, by 15.3% and 10.5%, respectively. Chinese kale grown under FR-3 had an 11.1% bigger stem diameter than CK. 

The leaf morphology of Chinese kale was altered by far-red light as well (Figure 1). Compared to CK, the leaf length (21.6% and 36.1%), leaf width (14.7% and 25.5%) and petiole length (19.6% and 50.1%) were promoted along with the increased far-red light intensity. The Chinese kale grown under FR-3 and FR-6 obtained a significantly enlarged leaf area (35.7% and 55.7%), while the nearly unchanged specific leaf area (SLA) implied an immutable leaf thickness. Additionally, the narrowed leaf angle caused by FR-3 and FR-6 suggested a more compact plant morphology.

Simultaneously, the plant biomass of Chinese kale greatly increased by supplementary far-red light (Table 1). The plant fresh weight of Chinese kale under FR-3 and FR-6 increased by 39.1% and 47.0%, respectively. Similarly, the increases in shoot fresh weight were 42.0% and 53.0%, respectively. The highest dry weights of plant and shoot were found under FR-3, with increases of 36.9% and 39.1%, followed by FR-6 (32.0% and 35.0%). The root mass and water content in each part of Chinese kale were almost invariable. Consequently, the root and shoot ratio significantly decreased under supplementary far-red light. 

### 2.2. Effects of Supplementary Far-Red Light on Photosynthetic Traits

Supplementary far-red light obviously affected the photosynthetic characteristics of Chinese kale (Table 2). The contents of chlorophyll a, chlorophyll b, total chlorophyll, and carotenoids increased by 71.33%, 77.33%, 72.37%, and 42.28%, respectively, under FR-3 and increased by 36.3%, 37.73%, 35.53%, and 30.00%, respectively, under FR-6. 

Regarding the chlorophyll fluorescence parameters (Table 2), Y(II) shared a similar trend with photosynthetic pigments. The Fv/Fm values of CK and far-red light treatments were all below 0.83. The NPQ, qL, and qP were promoted by FR-3 and FR-6, while Y(NPQ) was reduced. The qN value decreased along with the higher dosage of supplementary far-red light, while ETR presented an opposite trend. However, Y(NO) was not influenced by far-red light.

### 2.3. Effects of Supplementary Far-Red Light on Mineral Profiles

Supplementary far-red light significantly influenced the concentration and accumulation of mineral elements in Chinese kale (Table 3). The determined minerals included six major elements (N, P, K, Ca, Mg, S) and two minor elements (Zn, Fe). The concentration of N (3.32% and 6.20%), P (7.69% and 17.15%), Zn (10.17% and 10.17%), and Fe (46.71% and 51.56%) decreased along with the increased dosage of supplementary far-red light. However, the Ca concentration increased by 12.53% under FR-6. 

The accumulation of minerals presented different trends under far-red light supplementation (Table 3). The amounts of N (44.92% and 52.66%), P (38.16% and 34.87%), K (50.85% and 56.25%), Mg (46.77% and 66.67%), S (38.57% and 62.34), and Zn (34.48% and 46.36%%) markedly increased in Chinese kale grown under FR-3 and FR-6. The amount of Ca increased drastically in FR-3 (83.21%), and significantly increased in FR-6 (56.85%). However, the amount of Fe appeared to be little affected by supplementary far-red light.

### 2.4. Identification of Differentially Expressed Genes in Response to Supplementary Far-Red Light

The cDNAs of the stem-tip tissue samples from CK, FR-3 and FR-6 (three biological replicates per treatment, nine libraries in total) were prepared and sequenced using Illumina HiSeq 2500 to obtain a comprehensive transcriptome of Chinese kale. In this project, 75.3 GB HQ clean reads were generated. The average GC content, Q20, and Q30 of the nine libraries were 46.79%, 98.08%, and 94.33%, respectively. The output statistics of the sequencing are presented in Appendix A. Since the entire genome sequence of Chinese kale was unknown, the HQ clean reads were assembled using Trinity software. After clustering, 145746 unigenes (mean length: 1356 nt) with an N50 of 1801 nt were obtained (Appendix A). These results demonstrated that the Illumina sequencing data were effective and qualified for further analyses.

To understand the functional annotation of the transcriptome in Chinese kale, all assembled unigenes were then submitted against seven public databases, using BLASTx. The results showed that a total of 133,129 (91.34%) unigenes were annotated in at least one database (Appendix A). Proteins having the highest sequence similarity with the unigenes were more easily annotated by Trembl (89.98%) and NR (88.76%) than other databases. The similarity distribution of the NR database corresponded to the known sequences *Brassica* species and showed the highest homology with *Brassica oleracea* (47.05%), *Brassica napus* (34.32%), and *Brassica rapa* (6.05%), suggesting that the annotation results were credible (Appendix A). 

Subsequently, DESeq2 software was used to identify differentially expressed genes (DEGs) from CK versus FR-3, CK versus FR-6, and FR-3 versus FR-6 (Figure 2). This study selected 1409 nonredundant DEGs (1658 DEGs in total) (Figure 2a). There were 260 downregulated genes and 195 upregulated genes in pairwise CK vs FR-3, 518 downregulated genes and 375 upregulated genes in pairwise CK vs FR-6, and 146 downregulated genes and 164 upregulated genes in pairwise FR-3 vs FR-6, respectively. The 1409 DEGs were clustered into eight profiles using STEM software, including three significant expression profiles (profiles 0, 1, and 7) (Figure 2b). Profile 0 included 306 DEGs that were downregulated along with the increased intensity of supplementary far-red light; profile 1 possessed 151 DEGs that were downregulated by far-red light as well, but the gene expressions showed no significance between FR-3 and FR-6; profile 7 contained 188 DEGs that were upregulated with the increase in far-red light intensity (Figure 2b). To confirm the accuracy and reproducibility of RNA-seq expression profiles, qRT-PCR was conducted to examine the expression levels of 12 genes that were randomly selected from DEGs (Figure 2c). The relative expression levels were analyzed with the 2^−ΔΔCT^ method, using *EF1* and *UBQ* as the reference genes. All 12 genes had similar expression patterns and Pearson’s correlation coefficients between the RNA-seq and qRT-PCR data were among the range of 0.9008 to 0.9994, indicating the reliability of the RNA-seq data.

In this study, functions of all DEGs were classified by GO (Appendix A) and KEGG (Figure 2d) assignments. Among these subcategories, the GO terms ‘chlorophyll binding’, ‘photosystem’, and ‘photosynthesis, light harvesting’ were the most significantly enriched in molecular function, cellular component, and biological process, respectively. In addition, GO terms related to the biosynthesis of glucosinolates and the glucosides were significantly enriched biological processes as well. Meanwhile, the results of KEGG pathway enrichment revealed that the DEGs were mostly enriched in ‘photosynthesis–antenna proteins’ (n = 32, 5.18%), ‘glucosinolate biosynthesis’ (n = 17, 2.75%), ‘circadian rhythm–plant’ (n = 27, 4.37%), and ‘photosynthesis’ (n = 13, 2.10%) (Figure 2d). Additionally, ‘plant hormone signal transduction’ (n = 38, 6.15%) was included in the top 20 enriched pathways (Figure 2b). These results implied that DEGs involved in photosynthesis, plant hormone biosynthesis and signal transduction, and plant circadian rhythm might be the ones most worthy of study.

### 2.5. Expression Patterns of DEGs of Functionally Enriched Pathways under Supplementary Far-Red Light

To clarify the molecular adaptability of Chinese kale to far-red light, the expression patterns of DEGs of key pathways were analyzed (Figure 3). A total of 60 DEGs were related to photosynthesis, among which 54 genes were downregulated by supplementary far-red light (Figure 3a,b). In photosynthesis–antenna protein pathway (ko00196), 31 genes that annotated light harvesting Chl a/b-binding protein complex I (*LHCA1/3*) and II (*LHCB1/2/4/5/6*) showed a decreased transcription. In photosynthesis pathway (ko00195), 12 genes involved in electron transport (*petE*), H+/Na+-transporting ATPase subunit beta (*ATPF0A*), and reaction center of PSI (*psaA.1*, *psaO*, *psaG*) and PSII (*psbQ*, *psbR*, *psbY*, *psbO*, *psbB*) were downregulated by supplementary far-red light. In porphyrin and chlorophyll metabolism pathway (ko00860), seven genes, encoding pheophorbidase (PPD), geranylgeranyl diphosphate reductase (*chlP*), magnesium–protoporphyrin IX monomethyl ester (oxidative) cyclase (*chlE*), and chlorophyllide a oxygenase (*CAO*), were downregulated; while four genes, encoding uroporphyrinogen decarboxylase (*hemE*), and uroporphyrin-III C-methyltransferase (*cobA*), and protoporphyrin/coproporphyrin ferrochelatase (*hemH*), were upregulated. In carbon fixation and photosynthetic organisms pathway (ko00710), four genes, encoding glyceraldehyde-3-phosphate dehydrogenase (*GAPA*), fructose–bisphosphate aldolase (*ALDO*), phosphoribulokinase (*PRK*), and phosphoenolpyruvate carboxylase (*ppc*), were downregulated, while one gene, encoding malate dehydrogenase (*NAD-ME2*), was upregulated. 

In pathways related to plant hormone biosynthesis (Figure 3c) and signal transduction (Figure 3d), the gene expression patterns seemed more complex. This study presents that ten DEGs were enriched in tryptophan metabolism (ko00380), among which four genes, encoding auxin biosynthesis enzymes, were downregulated by supplementary far-red light, including N-hydroxythioamide S-beta-glucosyltransferase (*UGT74B1*), indole-3-acetaldehyde oxidase (*AAO1.2*), and indoleacetaldoxime dehydratase (*CYP71A13*) (Figure 3c). In the meantime, 10 genes were annotated as early response genes in the auxin signal transduction pathway, such as auxin response factors (*ARFs*), auxin-responsive protein IAA (*AUX/IAA*), auxin influx carrier (*AUX*), auxin responsive GH3 gene family (*GH3*), and SAUR family protein (*SAURs*). The upregulated genes included *ARF2/18*, *IAA26*, *GH3*, and *SUAR72*, while the downregulated genes included *ARF19*, *AUX1*, *SUAR15A*, and *SUAR50* (Figure 3d). The DEGs enriched in diterpenoid biosynthesis (ko00904), included genes encoding the key enzymes 3-beta-dioxygenase (*GA3ox*) and 2-oxoglutarate-dependent dioxygenase (*GA20ox1*) in GA biosynthesis, and gibberellin 2-oxidase (*GA2ox1*, *GA2ox2*) which turns the bioactive GAs into inactive GAs (Figure 3c). Meanwhile, three genes encoding GA signal-transduction-related TFs (*PIF3.2*, *PIF4*) were downregulated (Figure 3d). In the zeatin biosynthesis pathway (ko00908) which produces CTKs, only one DEG, cytokinin-N-glucosyltransferase (*UGT76C1/2*), was upregulated (Figure 3c). In the CTK signal transduction pathway, five response regulators (*ARRs*) were downregulated (*A-ARR4/6/8*, *B-ARR.1/2*) and 1 *ARR* was upregulated (*B-ARR.3*) (Figure 3d). This study also showed that six DEGs were enriched in the alpha-linolenic acid metabolism pathway (ko00592) which generates JAs (Figure 3c). The genes encoding OPC-8:0 CoA ligase 1 (*OPCL1*), acyl-CoA oxidase (*ACX*), enoyl-CoA hydratase (*MFP2*), and lipoxygenase (*LOX2S*) were upregulated, while acetyl-CoA acyltransferase 1 (*ACCA1*) and alpha-dioxygenase (*DOX*) were downregulated. As for JA signal transduction pathways, the jasmonate ZIM-domain-containing protein (*JAZ*) and transcription factor MYC2 (*MYC2*.1) were downregulated by supplementary far-red light (Figure 3d). 

In this study, 26 DEGs were enriched in the pathway of circadian rhythm–plant (ko04712) (Figure 3e), including 15 upregulated genes—phytochrome A (*PHYA*), phytochrome B (*PHYB*), two-component response regulator-like APRR1 (*TOC1*), APRR5 (*PRR5*), APRR7 (*PRR7.1/2*), protein TERMINAL FLOWER 1 (*FT*), and zinc-finger protein CONSTANS-LIKE (*CO9/10/15*)—and 10 downregulated genes: APRR7 (*PRR7.3*), CO4/5, REVEILLE 6-like (*LHY*), protein SPA1-RELATED 3-like (*COP1*), SUPPRESSOR OF PHYA-105 1-like (*SPA1*), and transcription factor PIF3-like (*PIF3.1*).

### 2.6. Endogenous Phytohormone Accumulation in Response to Supplementary Far-Red Light

To study the roles of endogenous phytohormones in the plant morphology of Chinese kale under supplementary far-red light, the contents of auxin, GAs, CTKs, and JAs were quantified (Figure 4). This study identified the active auxin indole-3-acetic acid (IAA), one of the naturally occurring auxin storage forms methyl indole-3-acetate (ME-IAA), and the oxidated auxin indole-3-carboxaldehyde (ICA) in Chinese kale (Figure 4a). Although the ICA content did not change, a significantly decreased IAA content and markedly increased ME-IAA content were observed under supplementary far-red light (Figure 4a). A total of seven types of GAs were detected, including two types of bioactive GAs (GA1, GA4) and five types of inactive GAs (GA_9_, GA_15_, GA_19_, GA_20_, GA_24_). The contents of GA_1_, GA_4_, and GA_24_ decreased significantly under supplementary far-red light, while the contents of GA_9_, GA_19_, and GA_20_ were obviously higher in far-red light treatments (Figure 4b). This study identified four types of CTKs, named N6-isopentenyladenine (IP), trans-zeatin (tZ), cis-zeatin (cZ), and dihydrozeatin (DZ). All CTKs presented significantly lower contents under supplementary far-red light (Figure 4c). As for JAs, methyl jasmonate (MEJA), jasmonic acid (JA), dihydrojasmonic acid (H2JA), and jasmonoyl-L-isoleucine (JA-ILE) were detected. Except for MEJA, which showed significantly lower content in FR-3 while higher content in FR-6, the other three JAs displayed markedly decreased contents under supplementary far-red light (Figure 4d).

### 2.7. Correlation Analysis of Phytohormones and Morphological Traits

To evaluate the potential relations between phytohormones and morphological traits and between minerals and morphological traits, Pearson’s correlation coefficient was introduced and exhibited with clustered heatmaps (Figure 5). 

As seen in Figure 5, 18 individual plant hormones were divided into two main clusters based on the positive and negative correlations with 22 morphological indices. The upper cluster included three types of GAs (GA_1_, GA_4_, GA_24_), two types of auxins (IAA, ICA), four types of CTKs (DZ, cZ, IP, tZ), and three types of JAs (JA, H2JA, JA-ILE). This cluster was characterized by negative correlations with the majority of morphological traits, namely biomass; width, length, and area of leaf; etc. However, GA_4_, GA_24_, cZ, IP, and JA-ILE showed significant positive correlations with leaf angle and root–shoot ratio. On the contrary, the lower cluster contained GA_9_, GA_19_, GA_20_, GA_15_, ME-IAA, and MEJA. This pattern was classified by positive correlations with most of the morphological traits. GA_9_, GA_19_, GA_20_, and ME-IAA were strongly and positively correlated with plant fresh weight, shoot fresh weight, plant height, internode length, leaf length, budding rate, leaf area, leaf width, and petiole length.

## 3. Discussion

Plants utilize light not only as energy source to survive but also as signal to regulate growth, development, and metabolism. Studies on changes of ambient light signals, which strongly influenced the biosynthesis and translocation of endogenous phytohormones, were rather complex due to plant species and organs. In our plant factory with negligible fluctuation of biotic and abiotic factors, this study generated growth and morphological indices, phytohormones profiles, and the transcriptome to uncover the response of Chinese kale to supplementary far-red light. Meanwhile, considering the particularity of far-red light, profiles of minerals were collected to define the regulatory mechanism.

### 3.1. The Elongation Growth of Chinese Kale under Supplementary Far-Red Light

Light drives plant growth and morphogenesis. The elongation growth of dicotyledonous species is stimulated by a low R:FR [4]. The plant height of the ornamental plants *Zinnia elegans* and *Cosmos bipinnatus* grown with a higher far-red light ratio (R:FR = 1.51) was 35% and 14%, respectively, which was significantly higher than that under a lower far-red light ratio (R:FR = 0.77) [31]. In this study, the far-red light treatments were applied to Chinese kale from the 15th day after sowing. At the 45th day of far-red light supplementation, the plant height was significantly promoted along with the enhanced far-red light dosage (Figure 1). Plant internodes sense the light environment and respond quickly to far-red light stimulation [32]. In this study, the internode length of Chinese kale showed a similar trend to plant height (Figure 1). Considering the unchanged leaf number, this study supposed that far-red light increased the plant height by promoting the elongation of internodes, rather than by higher internode numbers. In addition, other elongation-growth-related indices such as leaf length, leaf width, leaf area, and petiole length were elevated by increasing dosages of supplementary far-red lights (Figure 1).

In auxin signal-response pathways, both Aux/IAAs (repressors) and ARFs (either activators or repressors) function as transcriptional regulators [33]. ARFs can specifically bind to the TGTCTC auxin-response element (AuxRE) in the promoters of primary/early auxin-response genes such as members of the *AUX/IAA*, *GH3*, and *SAUR* families [34]. High levels of auxin lead to the ubiquitination and degradation of Aux/IAAs, release of ARFs and promotion of the expression of downstream auxin-responsive genes, while low levels of auxin allow the combination of Aux/IAAs and ARFs which suppress the expression of auxin-responsive genes [35]. In this study, the upregulated *AFR2* and *ARF18*, and the downregulated *AUX1* might promote the transcription of the auxin-responsive genes *GH3*, *SAUR72.1*, and *SAUR72.2* which consequently positively regulate the elongation growth of stem/internode and leaf of Chinese kale (Figure 3). 

A low R:FR ratio transformed active PHYB into the inactive form, allowing PIFs to induce shoot elongation in an auxin-dependent manner [36]. In parallel, increased GA levels contributed to the suppression of DELLA proteins (suppressors of PIFs in GA signal pathways), also resulting in shoot elongation [36]. Auxin is involved in in many developmental processes of plants in response to light stimuli, from gametogenesis, via embryogenesis, seedling growth, vascular patterning, and flower development, to senescence, functioning as a life-long steward [37]. The GAs comprise a large group of diterpenoid compounds, being widely used as plant growth regulators in agricultural production for their roles in promoting seed germination, stem elongation, and fruit development, and improving plant resistance to inversion [38]. The most common active GAs are GA_1_, GA_3_, and GA_4_, among which GA_1_ and GA_4_ universally occur in plants. The physiological functions of bioactive GAs can be generalized as stimulating organ growth through enhancement of cell elongation and, in some cases, cell division [39]. A low R:FR ratio, which induced high levels of GA_1_ and IAA in sunflower internodes, supports the causal growth-effectors in this process [14]. 

Our transcriptional data showed that *PHYA* and *PHYB* were upregulated while *PIF3/4/5* and five genes (*ACAT.2*, *AAO1.2*, *UGT74B1*, *CYP71A13.1/2*) of the auxin biosynthesis pathway were downregulated by supplementary far-red light (Figure 3). Meanwhile, the IAA content was reduced (Figure 4). These results indicated that the far-red light dosages (R:FR_FR-3_ = 12.76, R:FR_FR-6_ = 7.92) in this study might be not sufficient to be considered as the case of R: FR ratios low enough (R:FR_Sun light_ ≈ 1.15, R:FR_SAS_ < 0.6) to inhibit the protein activity of *PHYs*, but enough to work as signals that induce PHY transcription, and the lower expression of *PIF3/4/5* under far-red light treatments might contribute to the downregulation of auxin biosynthesis genes and consequently the decreased IAA level. However, the level of ME-IAA, the auxin precursor/storage form, which was predominantly abundant (326~690 times higher than IAA) accumulated with the increased dosage of supplementary far-red light (Figure 4a). In Arabidopsis, IAA CARBOXYMETHYLTRANSFERASE1 (IAMT1) converts IAA into ME-IAA, a non-polar methyl-esterified auxin that is presumed to be capable of transporter-independent movement [40]. The conversion from ME-IAA to IAA can be achieved by METHYL ESTERASE17 (MES17) [41]. Roles for MeIAA-derived auxin have been described in various aspects of development; the extent to which Me-IAA contributes to auxin homeostasis is not yet known [42]. 

On the other hand, in this study the levels of bioactive GA_1_ and GA_4_ decreased, but GA_9_ (precursor of GA_4_), and GA_19_ and GA_20_ (precursors of GA_1_) significantly accumulated under supplementary far-red light (Figure 4b), supporting the fact that the biosynthetic precursors and metabolites are often present at much higher concentrations than the bioactive hormones themselves [39]. Importantly, the correlation heatmap illustrated that the contents of ME-IAA, GA_9_, GA_19_, and GA_20_ had significant positive relations to plan height and internode length (Figure 5). Simultaneously, the leaf development, such as petiole length, and length, width, and area of leaf, were observed significantly positively correlated with the contents of ME-IAA, GA_9_, GA_19_, and GA_20_ (Figure 5). Additionally, the contents of these four hormones were found strongly and negatively correlated with the leaf angle while the contents of five other hormones (GA_4_, GA_24_, cZ, IP, JA, JA-ILE), which were decreased by far-red light, showed strongly positive correlations (Figure 5), suggesting that the function of these hormones is narrowing the angle between stem and leaf. Since a narrowed leaf angle allows Chinese kale to tolerate a high planting density in face of pressure on arable land, far-red light supplementation could be a useful tool to improve crop yields per unit land area and to gain a better understanding for high planting density breeding [2,43]. 

These results provided valuable insight into the regulatory mechanisms of the elongation growth or vegetative architecture of Chinese kale under supplementary far-red light by commonly recognized inactive, or precursors of auxin and gibberellins. 

### 3.2. The Biomass of Chinese Kale under Supplementary Far-Red Light

Photosynthesis is pivotal for the growth, yield, and quality of crops. Far-red light could change the plant biomass and chlorophyll contents [44]. A decreased R:FR (R:FR = 7.4, 1.2, and 0.8) resulted in a significantly decreased Chl a content, but increased the fresh weight and dry weight of tomato seedlings, while it slightly influenced the contents of Chl a and Chl b and the biomass of phyb mutants [45]. End-of-day (1 h) and diurnal (16 h) far-red light significantly increased the fresh weight (28.7% and 52.74%) and dry weight (24.98% and 39.22%) of the whole plant of lettuce, while it markedly decreased the content of total chlorophyll (12.44% and 31.84%) [27]. In this study, the contents of photosynthetic pigments (Chl a, Chl b, total Chl, and Car) were obviously increased by FR-3 (42.3%~77.3%) and FR-6 (30.0%~37.7%) (Table 2). Meanwhile, the fresh biomass (39.8%~53.0%) and dry biomass (32.0%~39.1%) of the whole plant and shoot of Chinese kale were noticeable elevated by supplementary far-red light (Figure 1), indicating that far-red light might promote the yield of Chinese kale through increasing photosynthetic pigment contents.

Furthermore, Fv/Fm is the maximum photochemical efficiency of PSII. Although the Fv/Fm values were all below 0.83, FR-3 and FR-6 helped Chinese kale obtain significantly higher Fv/Fm values (Table 2), suggesting that supplementary far-red light might to some degree alleviate the light stress in plant factories. Y(II) is the effective quantum yield of photosystem (PS II). FR-3 and FR-6 increased the Y(II) values by 24.24% and 12.15% (Table 2), respectively, showing a similar trend as photosynthetic pigment contents. This far-red-light-induced Y(II) in Chinese kale was consistent with that in lettuce [23]. Y(NPQ) is the regulated thermal-energy dissipation that protects plants from high light intensities. The reduced Y(NPQ) values indicated that supplementary far-red light helped Chinese kale to better adapt to the ambient light in the plant factory (Table 2). The qL and qP are photochemical quenching and the photochemical quenching coefficient, reflecting the fraction of open PS II centers. The increased qL and qP indicated that the PS II centers were activated by supplementary far-red light (Table 2). Additionally, the higher qP suggested a more effective photochemical electron transport [46]. The increased ETR by FR-3 and FR-6 also supported that (Table 2). Therefore, the supplementary far-red light might be effective in promoting the photosynthesis of Chinese kale.

Moreover, the downregulated key-enzyme encoding genes that help harvesting light (*LHCA1/3*, *LHCB1/2/4/5/6*) and benefit carbon-fixation (*PRK*, *ppc*, *ALDO*, *GAPA*) (Figure 3), indicated that far-red light participated in the reprogramming of photosynthesis-related pathways. Additionally, the increased leaf area and narrowed leaf angle allowed Chinese kale to obtain more light energy, which to some extent benefited the photosynthesis (Figure 1). The result of photosynthetic pigments, chlorophyll fluorescence parameters, leaf area and leaf angle, and the transcriptome data, all suggested that supplementary far-red light strongly influenced the photosynthetic-related processes which eventually contributed to the increased yield.

Furthermore, the functionally enriched plant circadian rhythm network was closely related to the photosynthesis-antenna protein pathway. In the plant circadian rhythm pathway, far-red light upregulated the expression of *PHYA*, which might have promoted the expression of *CO* (*CO9/10/15*) and the downstream gene *FT* through suppressing the activity of COP1–SPA1 complexes (Figure 3), and consequently led to higher budding rate (early flowering) of Chinese kale (Figure 1). These results imply that far-red light accelerated the transformation from vegetative growth to reproductive growth in a circadian rhythm-dependent manner. Additionally, the accelerated growth process was mainly reflected by higher biomass and elongated/expanded plant morphology of Chinese kale.

From the perspective of mineral nutrient uptake and utilization, far-red light could promote the sodium (Na) accumulation in tomato fruits, while it did not affect the accumulation of K, Mg, and Ca [47]. However, in this study, except for Fe, the determined mineral elements showed significantly increased amounts in Chinese kale treated with supplementary far-red light (Table 3). Since N, Mg, P, K, and S participate in components of formation of certain plant hormones, chlorophyll biosynthesis, help catalyze several key enzymes of photosynthesis, and are involved in the transport of carbohydrates and electrons [48], the increased amounts of N, P, K, Mg, and S by far-red light might contribute to the elevated contents of photosynthetic pigments, promote photosynthesis, help produce carbohydrates, and consequently increase the biomass. In addition, far-red lights had significant dose effects on Ca accumulation (Table 3), suggesting that Ca might act as a signal messenger to transmit far-red light signals and then regulate plant growth and development. 

A recent study reported that red, blue, and ultraviolet B light could regulate the uptake and utilization of minerals through photoreceptors and their downstream target genes [30]. In this study, *PHYA* and *PHYB* could be upregulated by FR-6 and FR-3, respectively (Figure 3e). Their target genes *PIF3/4/5* were downregulated by supplementary far-red light (Figure 3d). Since PIF4/5 directly regulate P-uptake-related and P-starvation-responsive genes [48], the P concentration of Chinese kale was reduced by supplementary far-red light (Table 3). *COP1.1* and *COP1.2* downregulated by FR-3 and FR-6 (Figure 3) might retard the formation of SPA1–COP1 complexes and therefore affect the uptake of N, K, and S. However, the elevated accumulation of mineral elements per Chinese kale plant enabled the primary requirements of fast growth and strong photosynthesis, even if the mineral concentration showed slight decreases (Table 3). 

In addition, the nearly unchanged moisture contents indicated that the far-red-light-elevated fresh weight of Chinese kale was not due to absorbing more water (Table 1). In addition, the decreasing root–shoot ratio along with the increasing far-red light dosages indicated that far-red light might drive the resources allocated to the shoot and thus improve the yield of Chinese kale (Figure 1). 

## 4. Materials and Methods

### 4.1. Plant Materials and Growth Conditions

This study was carried out in a plant factory with artificial light, South China Agriculture University (East longitude 113.36°, north latitude 23.16°). Seeds of Chinese kale (*Brassica alboglabra* Bailey cv. ‘Lvbao’) were sowed in moist sponge blocks (2 cm × 2 cm × 2 cm) and kept in a dark germination chamber for 48 h. The germinated seeds with sponge blocks were placed in a deep-flow-technique system with 1/2 strength of nutrient solution. The temperature (21 ± 2 °C), CO_2_ concentration (400–600 μmol·mol^−1^), relative humidity (55%–60%), and white LED light (250 μmol·m^−2^·s^−1^ PPFD, 8:00 to 18:00) were maintained accurately during cultivation. After 13 days, the seedlings with three true leaves were transplanted into a planting plate (90 cm × 60 cm, 24 plants/plate) with full strength of the nutrient solution. The full-strength nutrient solution (EC ≈ 2.0 mS·cm^−1^ and pH ≈ 6.0) was composed of the following elements: 56.0 mg·L^−1^ N, 22.8 mg·L^−1^ P, 184.7 mg·L^−1^ K, 80.0 mg·L^−1^ Ca, 24.0 mg·L^−1^ Mg, 64 mg·L^−1^ S, 5.6 mg·L^−1^ Fe, 0.5 mg·L^−1^ B, 0.5 mg·L^−1^ Mn, 0.05 mg·L^−1^ Zn, 0.02 mg·L^−1^ Cu, and 0.01 mg·L^−1^ Mo.

### 4.2. Light Treatments and Sample Preparation

Forty-eight hours after being transplanted, some plants were treated with supplementary far-red light at 3 W·m^−2^ (FR-3) and 6 W·m^−2^ (FR-6). The plants grown under white light were used as control (CK). The light spectra and details of light treatments are presented in Figure 6a and Table 4. During the experiment, other environmental conditions were set as similar to those at the seedling stage (Figure 6b). There were 72 plants per treatment in total. 

At the end of two photoperiods, samples of Chinese kale stem-tip tissue free of any mechanical damage were collected for RNA extraction. Five days later, Chinese kale stem-tip tissues were sampled for plant hormone quantification. There were 3 replicates per treatment, each replicate contained 24 plants. The samples were immediately frozen in liquid nitrogen, then stored at −80 °C until needed. After 45 days of light treatments, the fully-grown plants were harvested for image capture, and morphological and physiological index assays.

### 4.3. Measurements of Plant Mass and Morphology

The morphological assay was conducted on 9 plants per treatment. For each plant, the leaf number was counted, the fresh and dry mass of plant, shoot and root were weighed using an electronic scale. The plants were oven-dried at 75 °C for 7 days before dry mass measurements. The plant height, plant display (the maximum distance between blades under overhead view), and internode length were measured with a straightedge. The stem diameter was measured using a vernier caliper. The petiole length, leaf length, leaf width, and leaf area of the sixth fully-expanded leaf were calculated using a leaf area meter. The specific leaf area (SLA) was calculated according to the equation: SLA = leaf area/leaf dry weight. The budding rate was recorded based on all plants in each treatment. SPSS 25.0 was used for statistical analysis and the results are displayed with Excel 2019.

### 4.4. Photosynthetic Trait Determination

Chlorophyll a (Chl a), chlorophyll b (Chl b), and carotenoids (Car) were measured colorimetrically [49]. Fresh leaf tissues (0.5 g) were extracted with 95% ethanol (8.0 mL, *v*:*v*) and measured at 645, 663, and 440 nm with a UV spectrophotometer. The contents were calculated as follows: Chl a (mg·g^−1^) = (12.70 × OD_663_ − 2.69 × OD_645_) × V/1000 W; Chl b (mg·g^−1^) = (22.88 × OD_645_ − 4.67 × OD_663_) × V/1000 W; Car (mg·g^−1^) = (4.70 × OD_440_ − 2.17 × OD_663_ − 5.45 × OD_645_) × V/1000 W.

The chlorophyll fluorescence parameters were measured with mini-PAM-2500 (WALZ) after 30 min of dark-adaptation. The chlorophyll fluorescence parameters included the maximum PS II quantum yield (Fv/Fm), the quantum yield of photochemical energy conversion in PS II (Y(II)), the electron transport rate (ETR), the quantum yield of non-regulated non-photochemical energy loss in PS II (Y(NO)), the quantum yield of regulated non-photochemical energy loss in PS II (Y(NPQ)), the non-photochemical quenching coefficient (qN), the non-photochemical quenching (NPQ), the quenching coefficient (qL), and the coefficient of photochemical quenching (qP).

### 4.5. Mineral Element Determination

Samples of oven-dried Chinese kale were weighed, ground to powder, and stored to measure mineral element contents. Total nitrogen (N), phosphorus (P), potassium (K), calcium (Ca), magnesium (Mg), sulfur (S), zinc (Zn), and iron (Fe) was measured by using an atomic absorption spectrophotometry method [50]. The amount of each mineral element per plant was calculated (for example): N (mg/plant) = N concentration (mg·g^−1^) × shoot DW (g). 

### 4.6. Quantification of Phytohormones and Gibberellins

Auxin (IAA, ME-IAA, and ICA), jasmonate (MEJA, JA, H2JA, and JA-ILE), cytokinin (IP, tZ, cZ, and DZ), and gibberellin (GA_1_, GA_3_, GA_4_, GA_7_, GA_9_, GA_15_, GA_19_, GA_20_, GA_24_, and GA_53_) contents were determined using MetWare (http://www.metware.cn/, accessed on 22 March 2020) based on the AB Sciex QTRAP 6500 LC-MS/MS platform. SPSS 25.0 was used for statistical analysis and the results are displayed with Excel 2019.

Sample extraction: For JAs, CTKs, and auxins, the plant samples (50 mg) were frozen in liquid nitrogen, ground into powder, and extracted with methanol:water:formic acid (15:4:1, *v*:*v*:*v*). The combined extracts were evaporated to dryness using a nitrogen gas stream, reconstituted in 80% methanol (*v*:*v*), and filtered (PTFE, 0.22 μm; Anpel) before LC–MS/MS analysis. For gibberellins, the plant samples were ground using a mixer mill (MM 400, Retsch, Germany) for 1 min at 30 Hz. After that, the powdered samples (200 mg) were extracted overnight at 4 °C with 1500 uL 70% (*v*:*v*) acetonitrile, and ultrasound-assisted extraction was carried out for 30 min at ambient temperature, followed by vortexing (15 s) and centrifugation (14,000 rpm for 10 min). The supernatants were collected in a volume of 1000 uL, then evaporated to dryness using a nitrogen gas stream at room temperature, reconstituted in 100 uL 80% (*v*:*v*) methanol, and diluted to 800 uL with water. The extracts were passed through an SPE cartridge (200 mg, 3 mL; CNW) and evaporated to dryness using a nitrogen gas stream at ambient temperature. Then, the extracts were reconstituted in 200 uL 80% (*v*:*v*) methanol and filtered (PTFE, 0.22 μm; Anpel) before LC–MS/MS analysis.

HPLC conditions: The phytohormone extracts were analyzed using an LC–ESI–MS/MS system (HPLC, Shim-pack UFLC SHIMADZU CBM30A system, http://www.shimadzu.com.cn/, accessed on 20 March 2022; MS, Applied Biosystems 6500 Triple Quadrupole, http://www.appliedbiosystems.com.cn/, accessed on 20 March 2022). The analytical conditions were as follows: column, Waters ACQUITY UPLC HSS T3 C18 (1.8 µm, 2.1 mm * 100 mm); solvent system, water (0.05% acetic acid):acetonitrile (0.05% acetic acid); gradient program, 95:5 (*v*:*v*) at 0 min, 95:5 (*v*:*v*) at 1 min, 5:95 (*v*:*v*) at 8 min, 5:95 (*v*:*v*) at 9 min, 95:5 (*v*:*v*) at 9 min, 95:5 (*v*:*v*) at 12 min; flow rate, 0.35 mL/min; temperature, 40 °C; injection volume, 2 μL. The effluent was alternatively connected to an ESI–triple quadrupole-linear ion trap (QTRAP)–MS.

ESI–triple quadrupole-linear ion trap (QTRAP)–MS conditions: An AB 6500 QTRAP LC/MS/MS system, equipped with an ESI Turbo Ion-Spray interface, operating in both positive and negative ion mode and controlled by Analyst 1.6 software (AB Sciex), was used. The ESI source operation parameters were as follows: ion source, turbo spray; source temperature 500 °C; ion spray voltage (IS) 4500 V; curtain gas (CUR), set at 35.0 psi; collision gas (CAD), medium. DP and CE for individual MRM transitions were performed with further DP and CE optimization. A specific set of MRM transitions was monitored for each period according to the plant hormones eluted within this period.

### 4.7. RNA Extraction, Sequencing, and De Novo Assembly

For sequencing, total RNA was obtained using AG RNAex Pro Reagent (Accurate Biotechnology Co., Ltd., Changsha, China). The integrity of the extracted RNA was measured on a 1.2% denaturing agarose gel and in an Agilent Bioanalyzer Model 2100 (Agilent Technologies, Santa Clara, CA, USA). The RNA samples with high purity were used to construct cDNA libraries. The quality and quantity of the libraries were verified using an Agilent Bioanalyzer Model 2100 and real-time RT-PCR, respectively. Subsequently, nine purified cDNA libraries were sequenced using Illumina HiSeqTM 2500 platform (Illumina, San Diego, CA, USA) by MetWare (http://www.metware.cn/, accessed on 20 March 2022).

Raw reads in fastq format were first processed with in-house Perl scripts. In this step, clean reads were obtained by removing adapter sequences, low quality reads containing >10% N base and/or >50% of base with Q ≤ 20. The copyright for Perl script belongs to Metware (http://www.metware.cn/, accessed on 20 March 2022). Meanwhile, Q20, Q30, and GC contents were calculated. Then, the ribosome reads were removed using Bowtie [51]. Subsequently, high-quality clean reads were de novo assembled into gene sets separately using Trinity [52]. All raw reads were deposited to the National Center for Biotechnology Information Sequence Reads Archive (SRA, http://trace.ncbi.nlm.nih.gov/Traces/sra, accessed upon publication) with bioproject accession number PRJNA942455 and SRA accession numbers SAMN33691637, SAMN33691638, SAMN33691639, SAMN33691640, SAMN33691641, SAMN33691642, SAMN33691643, SAMN33691644, SAMN33691645.

### 4.8. Transcriptome Data Analysis

Transcripts assembled using Trinity were the number of unigenes. The unigenes were used for BLASTX alignment and annotation against seven public databases (KEGG, NR, SwissProt, Trembl, KOG, GO, and Pfam). The expression of unigenes was calculated based on their FPKM (fragments per kilobase of transcript per million mapped fragments) values with K-means method. The identification of differentially expressed genes (DEGs) was performed using DESeq2 [53]. In this study, a false discovery rate (FDR) < 0.05 and an absolute value of |log2 (Fold change)| > 0.585 were used as the threshold to determine significant DEGs.

To identify the putative biological functions and pathways for the DEGs (differentially expressed genes), GO (gene ontology) and KEGG (Kyoto encyclopedia of genes and genomes) enrichment analyses were conducted using OmicShare Tools (https://www.omicshare.com/tools/, accessed on 20 March 2022). In addition, GO and KEGG pathways with a q-value < 0.05 were significantly enriched in DEGs.

### 4.9. Validation of RNA-seq Data Using qRT-PCR

To verify the RNA-seq results, 12 unigenes were randomly selected for qRT-PCR analysis using *UBQ* and *EF1* as internal reference genes. The reactions were performed in a LightCycler 480 system (Roche, Basel, Switzerland) with an Evo M-MLV RT-PCR kit (Accurate Biotechnology Co., Ltd., Changsha, China), in accordance with the manufacturer’s instructions. The amplification was carried out with the following cycling parameters: the denaturation at 95 °C for 30 s, followed by 40 cycles of amplification at 95 °C for 5 s, and annealing at 60 °C for 30 s. The melting curves were analyzed at the end of 40 cycles (95 °C for 5 s, followed by a constant increase from 60 to 95 °C). Relative gene-expression levels were normalized to the mean of *UBQ* and *EF1* and calculated according to the 2^−ΔΔCT^ method [54]. To ensure the reproducibility of results, qPCR reactions for RNA-seq validation were prepared in three biological replicates. SPSS 25.0 was used for statistical analysis and the results are displayed with Excel 2019.

## 5. Conclusions

This study emphasized that supplementary far-red light in a plant factory could simultaneously promote the vegetative growth and accelerate the reproductive growth of Chinese kale through regulating expression of key genes in plant circadian rhythm pathway; elevate the elongation growth and modify the vegetative architecture of Chinese kale in a phytohormone-dependent manner; and increase the plant biomass of Chinese kale by enhanced photosynthesis, and higher accumulation of photosynthetic pigments and mineral elements. These results provide strong evidence that far-red light participated in morphological changes and phytohormone and transcriptome reprogramming of Chinese kale (summarized in Figure 7). In particular, this study strongly suggests applying far-red light supplementation at 6 W·m^−2^ in plant factory systems for the purpose of higher density cultivation, higher yield per unit area, quicker growth, and more crop rounds of vegetable production.

## Figures and Tables

**Figure 1 ijms-24-05563-f001:**
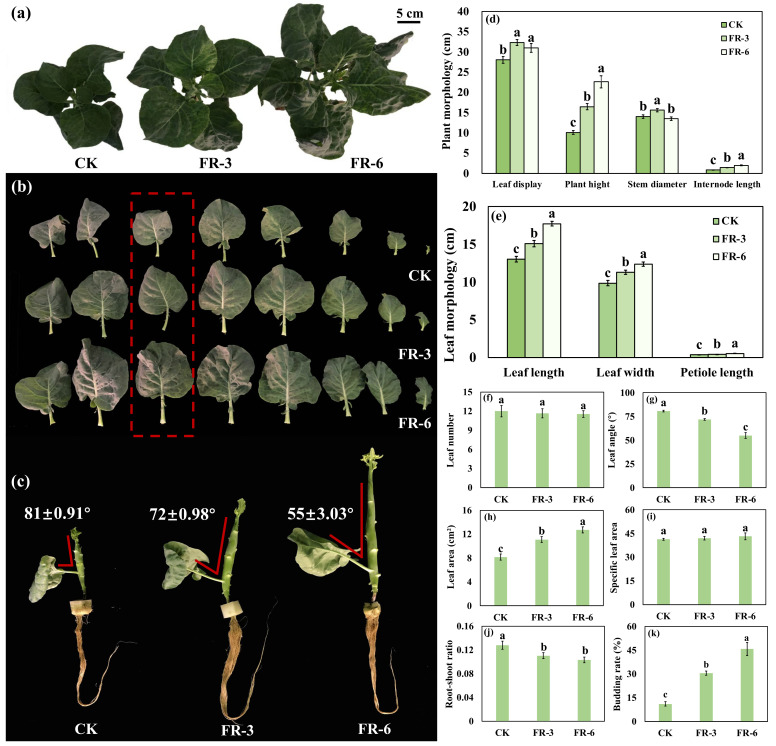
Morphological indices of Chinese kale (*Brassica alboglabra* Bailey) under supplementary far-red light. Scale bars in (**a**–**c**) are 5 cm. The (**d**) plant morphology, (**e**) leaf morphology, (**f**) leaf number, (**g**) leaf angle, (**h**) leaf area, (**i**) specific leaf area, (**j**) root-shoot ratio, and (**k**) budding rate of Chinese kale. The 6th true leaf (red dotted box) was used for leaf morphological analysis. Data are presented as mean ± standard deviations. Different letters indicate significant differences according to Tukey’s honest significant difference tests (*p* ≤ 0.05). CK, FR-3, and FR-6 indicate far-red light supplementation at 0 W·m^−2^, 3 W·m^−2^, and 6 W·m^−2^, respectively.

**Figure 2 ijms-24-05563-f002:**
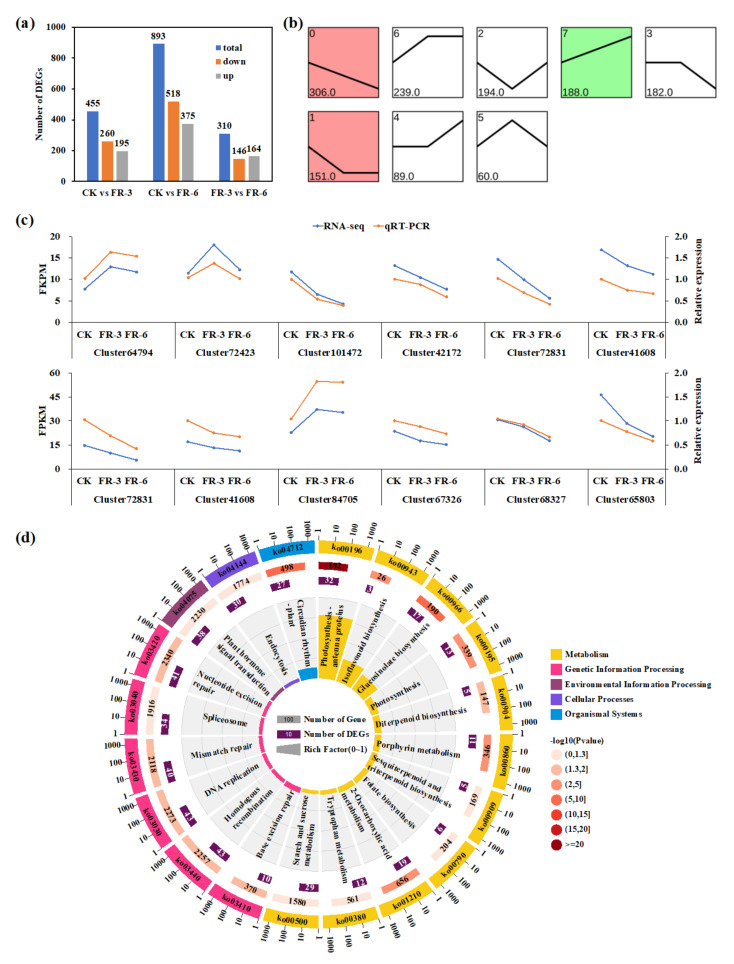
Analyses conducted for differentially expressed genes of Chinese kale (*Brassica alboglabra* Bailey) grown under supplementary far-red light. (**a**) Pairwise comparisons of gene expression; (**b**) Gene-expression patterns in the eight major profiles; the colored profile indicates their statistical significance (*p* ≤ 0.05) (red, up-regulated; green, down-regulated); (**c**) qRT-PCR validation of 12 DEGs identified using RNA-seq; (**d**) The top 20 pathways enriched from functional analysis. CK, FR-3, and FR-6 indicate far-red light supplementation at 0 W·m^−2^, 3 W·m^−2^, and 6 W·m^−2^, respectively.

**Figure 3 ijms-24-05563-f003:**
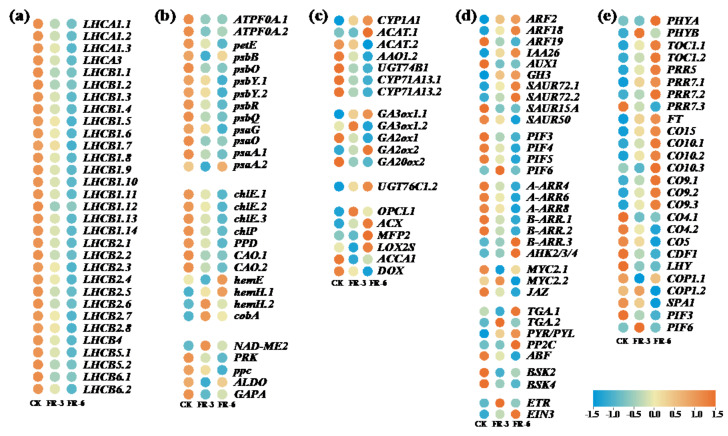
Expression analyses conducted for differentially expressed genes of Chinese kale (*Brassica alboglabra* Bailey) grown under supplementary far-red light. (**a**) Photosynthesis–antenna protein pathway (ko00196); (**b**) Pathways of photosynthesis (ko00195), porphyrin and chlorophyll metabolism (ko00860), and carbon fixation in photosynthetic organisms (ko00710); (**c**) Biosynthetic pathways related to auxin (ko00380), gibberellins (ko00904), cytokinins (ko00908), and jasmonates (ko00592); (**d**) Plant hormone signal transduction pathway (ko04075); (**e**) Plant circadian rhythm pathway (ko04712). Values from −1.5 to 1.5 indicate expression fold-change. CK, FR-3, and FR-6 indicate far-red light supplementation at 0 W·m^−2^, 3 W·m^−2^, and 6 W·m^−2^, respectively.

**Figure 4 ijms-24-05563-f004:**
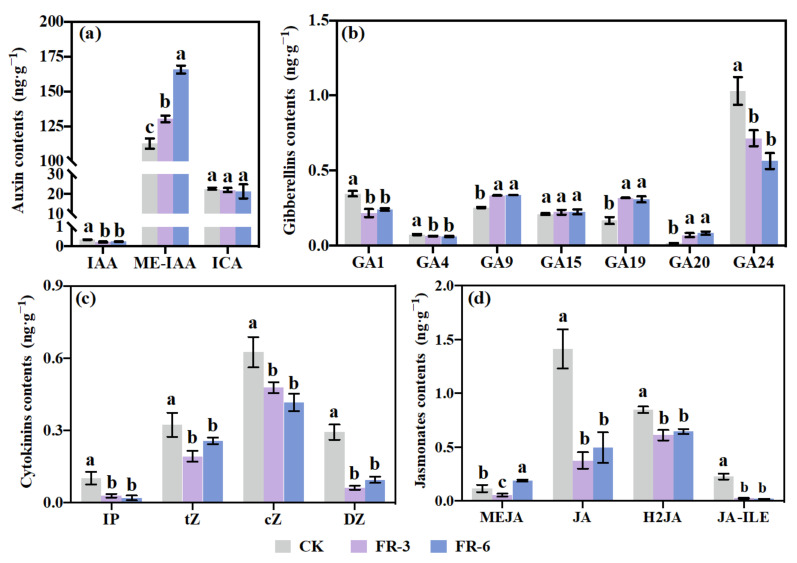
The contents of endogenous phytohormones of Chinese kale (*Brassica alboglabra* Bailey) grown under supplementary far-red light. Composition and contents of (**a**) auxins; (**b**) gibberellins, (**c**) cytokinins, and (**d**) jasmonates. Different letters in each bar plot indicated significant differences calculated using one-way ANOVA followed by Tukey’s honest significant difference tests (*p* ≤ 0.05). CK, FR-3, and FR-6 indicate far-red light supplementation at 0 W·m^−2^, 3 W·m^−2^, and 6 W·m^−2^, respectively. IAA, indole-3-acetic acid; ME-IAA, methyl indole-3-acetate; ICA, indole-3-carboxaldehyde; GA, gibberellin; IP, N6-isopentenyladenine; tZ, trans-zeatin; cZ, cis-zeatin; DZ, dihydrozeatin; MEJA, methyl jasmonate; JA, jasmonic acid; H2JA, dihydrojasmonic acid; JA-ILE, jasmonoyl-L-isoleucine.

**Figure 5 ijms-24-05563-f005:**
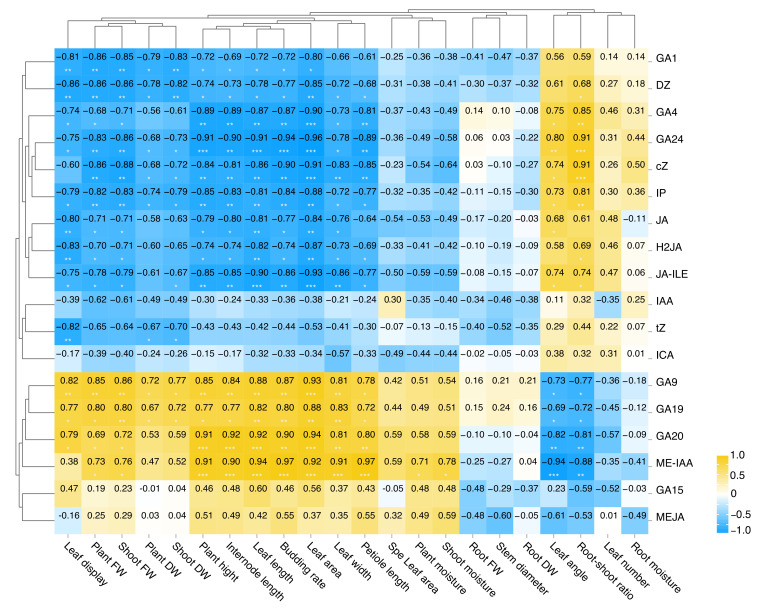
The correlation heatmap illustrates the relations between endogenous phytohormones and morphological indices of Chinese kale (*Brassica alboglabra* Bailey) grown under supplementary far-red light. The color from blue to yellow indicates correlation from negative to positive. The numbers (−1 to 1) show Pearson’s correlation coefficients between column and row. The asterisks indicate significance: *, **, and *** represent significances at 0.01 < *p* < 0.05, 0.001 < *p* < 0.01, and *p* ≤ 0.001, respectively, tested using Pearson’s two-tailed method. IAA, indole-3-acetic acid; ME-IAA, methyl indole-3-acetate; ICA, indole-3-carboxaldehyde; GA, gibberellin; IP, N6-isopentenyladenine; tZ, trans-zeatin; cZ, cis-zeatin; DZ, dihydrozeatin; MEJA, methyl jasmonate; JA, jasmonic acid; H2JA, dihydrojasmonic acid; JA-ILE, jasmonoyl-L-isoleucine.

**Figure 6 ijms-24-05563-f006:**
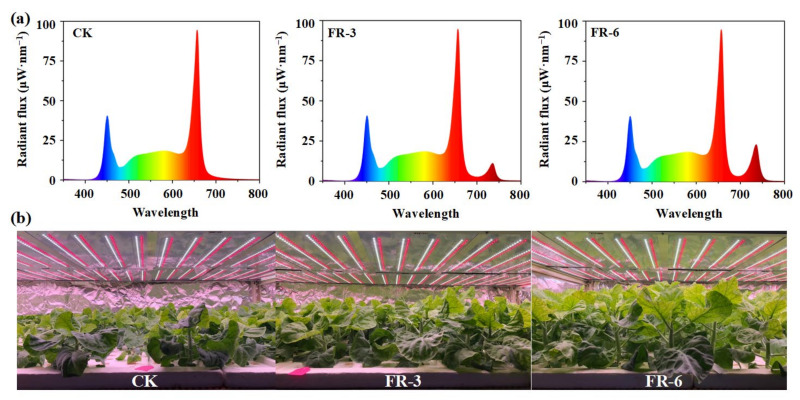
Light spectrum and plant growth conditions. (**a**) The light spectrum of the control group and far-red light treatments; (**b**) The growth conditions of Chinese kale at the 45th day after light treatments. CK, FR-3, and FR-6 indicate far-red light supplementation at 0 W·m^−2^, 3 W·m^−2^, and 6 W·m^−2^, respectively.

**Figure 7 ijms-24-05563-f007:**
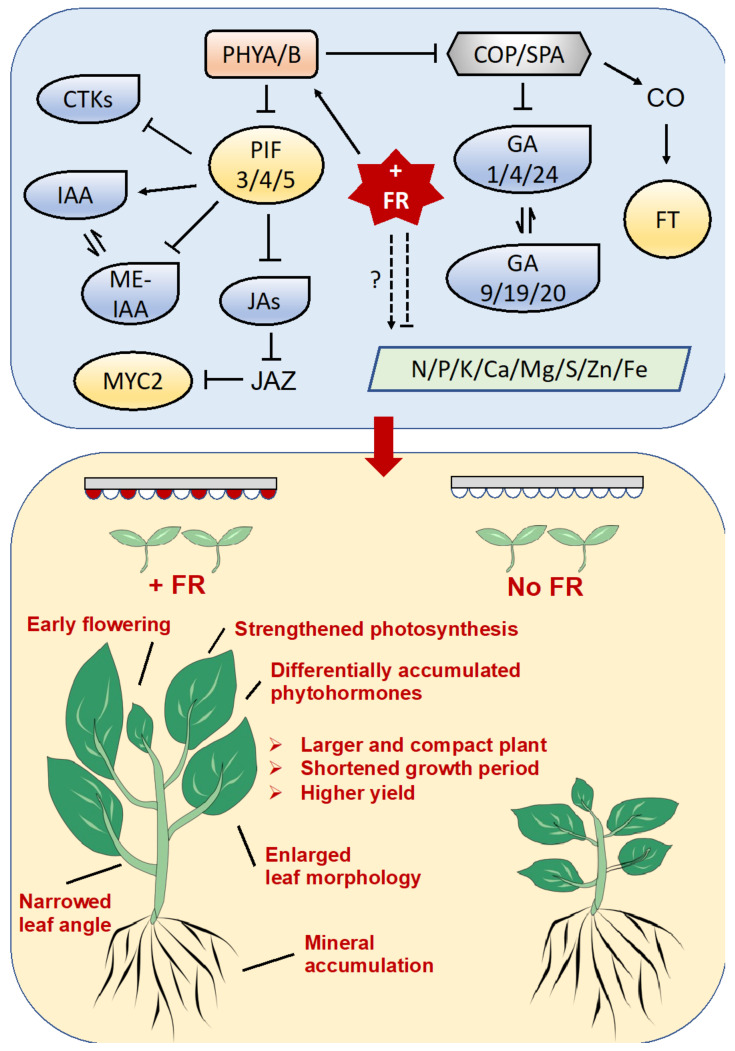
The far-red light (FR)-induced morphology changes, phytohormones, and transcriptome reprogramming of Chinese kale. In plant factories, supplementary far-red light simultaneously promoted the vegetative growth and accelerate the reproductive growth of Chinese kale through regulating expression of key genes in plant circadian rhythm pathways, elevate the elongation growth and modify the vegetative architecture of Chinese kale in a phytohormone-dependent manner, and increase the plant biomass of Chinese kale by enhanced photosynthesis, as well as higher accumulation of photosynthetic pigments and mineral elements. PHYA/B, phytochrome A and B; PIFs, phytochrome-interacting factors; JAZ, the jasmonate ZIM domain-containing protein; MYC2, bHLH family transcription factor; COP/SPA, suppressor of phytochrome a1–constitutively photomorphogenic1 (cop1)–suppressor complex; CO, protein CONSTANS; FT, protein TERMINAL FLOWER 1; IAA, indole-3-acetic acid; ME-IAA, methyl indole-3-acetate; ICA, indole-3-carboxaldehyde; GA, gibberellin; IP, N6-isopentenyladenine; tZ, trans-zeatin; cZ, cis-zeatin; DZ, dihydrozeatin; MEJA, methyl jasmonate; JA, jasmonic acid; H2JA, dihydrojasmonic acid; JA-ILE, jasmonoyl-L-isoleucine. N, nitrogen; P, phosphorus; K, potassium; Ca, calcium; Mg, magnesium; S, sulfur; Zn, zinc; and Fe, iron.

**Table 1 ijms-24-05563-t001:** Biomass indices of Chinese kale (*Brassica alboglabra* Bailey) under supplementary far-red light.

Biomass Indices	Light Treatments		
	CK	FR-3	FR-6
Fresh weight (g)			
Plant	55.94 ± 4.31 b	78.22 ± 2.92 a	82.29 ± 4.11 a
Shoot	50.23 ± 3.87 b	71.32 ± 2.83 a	76.69 ± 3.92 a
Root	5.72 ± 0.51 b	6.90 ± 0.36 a	5.60 ± 0.23 b
Dry weight (g)			
Plant	3.59 ± 0.34 b	4.91 ± 0.25 a	4.75 ± 0.18 a
Shoot	3.18 ± 0.30 b	4.43 ± 0.23 a	4.30 ± 0.16 a
Root	0.41 ± 0.05 a	0.49 ± 0.03 a	0.44 ± 0.03 a
Moisture content (%)			
Plant	93.62 ± 0.16 a	93.73 ± 0.16 a	94.18 ± 0.23 a
Shoot	93.71 ± 0.13 b	93.81 ± 0.14 b	94.34 ± 0.21 a
Root	92.80 ± 0.50 a	92.86 ± 0.46 a	91.97 ± 0.60 a

Data are presented as mean ± standard deviations. Different letters in each row indicated significant differences calculated using one-way ANOVA followed by Tukey’s honest significant difference tests (*p* ≤ 0.05). CK, FR-3, and FR-6 indicate far-red light supplementation at 0 W·m^−2^, 3 W·m^−2^, and 6 W·m^−2^, respectively.

**Table 2 ijms-24-05563-t002:** Photosynthetic traits of Chinese kale (*Brassica alboglabra* Bailey) under supplementary far-red light.

Photosynthetic Traits	Light Treatments		
	CK	FR-3	FR-6
Photosynthetic pigments (mg·g^−1^)			
Chlorophyll a	0.55 ± 0.01 c	0.95 ± 0.06 a	0.75 ± 0.03 b
Chlorophyll b	0.20 ± 0.00 c	0.35 ± 0.03 a	0.27 ± 0.01 b
Total chlorophylls	0.76 ± 0.01 c	1.31 ± 0.09 a	1.03 ± 0.04 b
Carotenoids	0.11 ± 0.00 c	0.16 ± 0.00 a	0.15 ± 0.00 b
Chlorophyll fluorescence parameters			
Fv/Fm	0.74 ± 0.01 b	0.76 ± 0.01 a	0.76 ± 0.01 a
Y(II)	0.33 ± 0.01 c	0.41 ± 0.02 a	0.37 ± 0.01 b
ETR	39.39 ± 1.05 c	43.90 ± 0.70 b	48.11 ± 0.73 a
Y(NO)	0.44 ± 0.01 a	0.44 ± 0.01 a	0.44 ± 0.01 a
Y(NPQ)	0.26 ± 0.02 a	0.21 ± 0.01 b	0.21 ± 0.01 b
qN	0.51 ± 0.00 a	0.46 ± 0.01 b	0.41 ± 0.01 c
NPQ	0.56 ± 0.00 b	0.60 ± 0.02 a	0.60 ± 0.01 a
qL	0.33 ± 0.01 b	0.36 ± 0.01 a	0.36 ± 0.01 a
qP	0.50 ± 0.01 b	0.60 ± 0.01 a	0.59 ± 0.00 a

Data are presented as mean ± standard deviations. Different letters in each row indicated significant differences calculated using one-way ANOVA followed by Tukey’s honest significant difference tests (*p* ≤ 0.05). CK, FR-3, and FR-6 indicate far-red light supplementation at 0 W·m^−2^, 3 W·m^−2^, and 6 W·m^−2^, respectively. Fv/Fm, is the maximum PS II quantum yield; Y(II), the quantum yield of photochemical energy conversion in PS II; ETR, the electron transport rates; Y(NO), the quantum yield of non-regulated non-photochemical energy loss in PS II; Y(NPQ), the quantum yield of regulated non-photochemical energy loss in PS II; qN, the non-photochemical quenching coefficient; NPQ, the non-photochemical quenching; qL, the quenching coefficient, a parameter estimating the fraction of open PS II centers based on a lake model; and qP, the coefficient of photochemical quenching, a parameter quantifying the photochemical capacity of photosystem II.

**Table 3 ijms-24-05563-t003:** The concentration and accumulation of mineral elements in Chinese kale (*Brassica alboglabra* Bailey) under supplementary far-red light.

Mineral Elements	Light Treatments		
	CK	FR-3	FR-6
Major element concentration (g·kg^−1^)			
N	47.74 ± 0.35 a	46.15 ± 0.68 ab	44.78 ± 0.67 b
P	5.51 ± 0.04 a	5.08 ± 0.06 b	4.56 ± 0.11 c
K	61.70 ± 0.67 ab	62.14 ± 0.73 a	59.36 ± 0.92 b
Ca	34.07 ± 0.44 b	35.62 ± 0.48 b	38.34 ± 0.79 a
Mg	6.75 ± 0.10 a	6.60 ± 0.13 a	6.90 ± 0.08 a
S	19.81 ± 0.21 a	18.33 ± 0.43 a	19.76 ± 0.83 a
Minor element concentration (mg·kg^−1^)			
Zn	33.85 ± 0.46 a	30.41 ± 0.85 b	30.41 ± 0.63 b
Fe	40.75 ± 6.95 a	21.72 ± 1.55 b	19.74 ± 1.65 b
Major element accumulation (g per plant)			
N	131.80 ± 17.11 b	191.00 ± 21.03 a	201.20 ± 16.49 a
P	15.20 ± 1.84 b	21.00 ± 1.99 a	20.50 ± 1.84 a
K	170.50 ± 23.09 b	257.20 ± 26.42 a	266.40 ± 16.87 a
Ca	94.10 ± 13.02 c	147.60 ± 18.07 b	172.40 ± 17.62 a
Mg	18.60 ± 2.30 b	27.30 ± 3.12 a	31.00 ± 2.42 a
S	54.70 ± 6.89 b	75.80 ± 8.14 a	88.80 ± 11.78 a
Minor element accumulation (mg per plant)			
Zn	0.09 ± 0.01 b	0.13 ± 0.01 a	0.14 ± 0.01 a
Fe	0.12 ± 0.06 a	0.09 ± 0.02 a	0.09 ± 0.02 a

Data are presented as mean ± standard deviations. Different letters in each row indicated significant differences calculated using one-way ANOVA followed by Tukey’s honest significant difference tests (*p* ≤ 0.05). CK, FR-3, and FR-6 indicate far-red light supplementation at 0 W·m^−2^, 3 W·m^−2^, and 6 W·m^−2^, respectively. N, nitrogen; P, phosphorus; K, potassium; Ca, calcium; Mg, magnesium; S, sulfur; Zn, zinc; and Fe, iron.

**Table 4 ijms-24-05563-t004:** Spectral data for the light-emitting diodes (LEDs).

Parameters	Light Treatments		
	CK	FR-3	FR-6
Single-band photon flux density (μmol·m^−2^·s^−1^)			
UV-A (350–400 nm)	0.27	0.35	0.13
B (400–500 nm)	31.41	29.66	29.93
G (500–600 nm)	47.92	46.51	47.00
R (600–700 nm)	170.73	168.96	167.06
FR (700–800 nm)	4.30	13.24	22.91
Integrated photon flux density (μmol·m^−2^·s^−1^)			
PPFD	250.06	245.13	243.99
YPFD	223.73	219.76	218.61
TPFD	224.71	222.07	222.20
Radiation ratio			
R/B	5.44	5.70	5.58
R/G	3.56	3.63	3.55
R/FR	39.68	12.76	7.29
Daily light integral (mol·m^−2^·d)			
10 h	9.00	8.82	8.78

CK, FR-3, and FR-6 indicate far-red light supplementation at 0 W·m^−2^, 3 W·m^−2^, and 6 W·m^−2^, respectively. UV-A, ultraviolet-A light; B, blue light; G, green light; R, red light; FR, far-red light. PPFD, photosynthetic photon flux density; YPFD, yield photon flux density; TPFD, total photon flux density.

## Data Availability

The research data are available from the corresponding author.

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
