# Peer review of "Far-Red-Light-Induced Morphology Changes, Phytohormone, and Transcriptome Reprogramming of Chinese Kale (Brassica alboglabra Bailey)"

_ijms, 2023, doi:10.3390/ijms24065563_

Round 1
Reviewer 1 Report
The manuscript describes a well designed and exhaustive experiment. Statistical and informatic tools are satusfactory. Results are robust and Discussion is adequate. Conclusions are supported by experimental results and data analysis. I suggest some changes in Introduction because in my opinion it is rather long and some information is unnecessary. Please consider to present in Introduction more concise ideas on the studied problem.
Author Response
Replies to the Reviewer1’s comments
Thank you for your recognition on our research. This manuscript has been revised carefully according to your comments and suggestions. All the revisions were clearly highlighted in red. The replies to your comments are listed as follows:
1) I suggest some changes in Introduction because in my opinion it is rather long and some information is unnecessary. Please consider to present in Introduction more concise ideas on the studied problem.
Answers: Thank you for making our manuscript more concise. We have duly condensed the Introduction according to your kind suggestion. Please see the revision in Introduction section.
Many thanks for your comments and suggestions.

Reviewer 2 Report
The study of Li et al. provided evidence that far-red light participated in morphology changes, phytohormone and transcriptome reprogramming of Chinese kale. An interesting finding; as the supplementary far-red light can be a useful tool to regulate the vegetative architecture, enhance the photosynthesis, benefit minerals accumulation, accelerate the growth, and obtain significant higher yield of Chinese kale.
The paper is well-presented and the English is clear and easy to read. The introduction is well structured and informative, the objectives are clear and well presented, the methods are well established and appropriated, the results are simple, well defined and easy to understand. The discussion is adequate. A very nice work.
Some minor corrections and suggestions need to be taken into account:
The title:
Line 3: you should add the scientific name of Chinese kale
Introduction section:
Line 68: when you talk about phytochrosms (PHYA, B/D/E) you should define the hole word (mentioned for the 1st time) to be easier for readers. Please verify in all the introduction.
Line 99-100: Define HY5 and put the hole words of the abbreviations used. I suggest you to mention that HY5, a key component of light signaling (See this paper of Nawkar et al. (2017) https://doi.org/10.1073/pnas.1609844114) and that HY5 gene is responsible for the regulation of fundamental developmental processes of the plant cell: cell elongation, cell proliferation, and chloroplast development.
Line 108: “co-analyzing “ instead of “co-analyzed »
Results Section:
In Figure 1: you should add in the title the meaning of CK, FR-3 and FR-6. Title of figures and tables should be explanatory without need to go to material & methods section to understand the abreviations used.
Table 1. you should add (Brassica alboglabra Bailey) after Chinese kale. and in the footnote of this Table 1 add the explanation of these lighting treatments CK, FR-3 and FR-6
Table 2: in the title add the latin name of chinese kale. In the foototes of this table, add the explanation of all the abrevaiated parameters of the Chlorophyll fluorescence parameters (Y (II), ETR, Y (NO), ...etc)
Line 174: The accumulation of minerals presented different trend with --- (it should be "supplementary far-red light" instead of "concentration" no?)
Line 202-203: Brassica oleracea, napus and rapa should be in itallic, please verify in all the document
Line 207: Please verify the value "1409" because in Figure 2-a I found there is 1658 DEGs.
Line 213: "far-red" instead of "fa-red"
Line 236: in the Title of figures avoid to put abreviations; it is better to use the hole words "differentially expressed genes" instead of "DEGs".
Same for Figure 3, Line 297.
Material & Methods section:
Lines 528-530: You should also add the (R: FR ratio).
Line 559: You did not give detail of the Chlorophyll fluorescence parameters that you evaluated, you should add the detail of each parameter that appears in the results section.
Line 624: you should add "Fragments per kilo base of transcript per million mapped fragments" after "FPKM"
Line 628: add the hole name of GO and KEGG that appears for the 1st time.
Conclusion section:
Figure 6: Line 666, in the title, you should add (FR) after "far-red light" and add at the end of the title, the explanation of the abbreviations used in the top figure.
Supplementary file:
Fig. S1 a: Enhance the quality of this Fig (a) the “x” and “y” axes are not lisible. You should also add in the title the meaning of each light treatment (CK, FR-3 and FR-6)
Fig. S2: You should enhance the quality the legend is not clear.
Table S4: You should add the webpage link of each protein database.
Author Response
Replies to the Reviewer2’s comments
Thank you for your time and efforts on our manuscript. This manuscript has been revised carefully according to your comments and suggestions. All the revisions were clearly highlighted in red. The replies to your comments are listed as follows:
[The Title]
- Line 3: you should add the scientific name of Chinese kale.
Answers: Thank you. We have added the scientific name of Chinese kale in the title. Please see the revised title in Line 2-4.
[Introduction Section]
- Line 68: when you talk about phytochrosms (PHYA, B/D/E) you should define the hole word (mentioned for the 1st time) to be easier for readers. Please verify in all the introduction.
Answers: Thank you for this kind suggestion. We have carefully checked the abbreviations and added the definition when mentioned for the 1st time. Please see the revision in Introduction section.
- Line 99-100: define HY5 and put the hole words of the abbreviations used. I suggest you to mention that HY5, a key component of light signaling (Nawkar et al. (2017) https://doi.org/10.1073/pnas.1609844114) and that HY5 gene is responsible for the regulation of fundamental developmental processes of the plant cell: cell elongation, cell proliferation, and chloroplast development.
Answers: Thank you making our manuscript more concise. The related literatures have been cited. Please see the revision in Line 85-87.
- Line 108: ‘co-analyzing’ instead of ‘co-analyzed’.
Answers: Thank you. We have corrected this word. Please see it in Line 96.
[Results Section]
1) In Figure 1: you should add in the title the meaning of CK, FR-3 and FR-6. Title of figures and tables should be explanatory without need to go to material & methods section to understand the abreviations used.
Answers: Thank you. We have added the scientific name of Chinese kale. Please see the revised title in Figure 1 (Line 131).
2) Table 1: you should add (Brassica alboglabra Bailey) after Chinese kale. and in the footnote of this Table 1 add the explanation of these lighting treatments CK, FR-3 and FR-6.
Answers: Thank you for this kind suggestion. We have added the scientific name of Chinese kale. And the explanations of lighting treatments were added in the footnotes of all figures and tables. Please see the revision in Result section.
3) Table 2: in the title add the latin name of Chinese kale. In the footnotes of this table, add the explanation of all the abbreviated parameters of the Chlorophyll fluorescence parameters (Y (II), ETR, Y (NO), ...etc).
Answers: Thank you. We have added the explanation of all the abbreviated parameters of the chlorophyll fluorescence parameters. Please see the information in Line 158-165.
4) Line 174: the accumulation of minerals presented different trend with --- (it should be "supplementary far-red light" instead of "concentration" no?)
Answers: We are grateful that you pointed out this problem. We have revised this sentence. Please see it in Line 173-174.
5) Line 202-203: Brassica oleracea, napus and rapa should be in itallic, please verify in all the document.
Answers: Thank you for the kind suggestion. We have verified the format of these words. Please see them in Line 203-204.
6) Line 207: please verify the value ‘1409’ because in Figure 2a I found there is 1658 DEGs.
Answers: Thank you for pointing out this. In three comparison pairs (CK vs FR-3, CK vs FR-6, FR-3 vs FR-6), there are 1658 DEGs in indeed. However, there were only 1409 DEGs after excluding the duplicate genes. We have added the explanation in this sentence. Please see it in Line 207-208.
7) Line 213: ‘far-red’ instead of ‘fa-red’.
Answers: Thank you. We have corrected this spelling. Please see it in Line 214.
8) Line 236: in the title of figures avoid to put abreviations; it is better to use the hole words ‘differentially expressed genes’ instead of ‘DEGs’.
Answers: Thank you. We have revised all titles according to your kind suggestion. Please see it in Line 237.
9) Line 297: same for Figure 3.Answers: Thank you. We have revised the title. Please see it in Line 299.
[Material & Methods section]
1) Lines 528-530: you should also add the (R: FR ratio).
Answers: Thank you. Since another reviewer suggested to move supplementary Table 1 (Table 4. Spectral data for the light-emitting diodes) to the main text, the detailed information could be seen clearly now. Please see the revision in Table 4 (Line 557).
2) Line 559: you did not give detail of the Chlorophyll fluorescence parameters that you evaluated. you should add the detail of each parameter that appears in the results section.
Answers: Thank you. The explanation of chlorophyll fluorescence parameters were added according to your kind suggestion. Please see the revision in Line 581-587.
3) Line 624: you should add ‘Fragments per kilo base of transcript per million mapped fragments’ after ‘FPKM’.
Answers: Thank you. We have added this definition. Please see it in Line 651.
4) Line 628: add the hole name of GO and KEGG that appears for the 1st time.
Answers: Thank you. We have added the explanations. Please see them in Line 656-658.
[Conclusion Section]
1) Figure 6: Line 666, in the title, you should add (FR) after ‘far-red light’ and add at the end of the title, the explanation of the abbreviations used in the top figure.
Answers: Thank you. We have carefully added the information in Figure 7 according to your suggestion. Please see the revision in Figure 7 (Line 687-702).
[Supplementary File]
1) Fig. S1 a: enhance the quality of this Fig (a) the “x” and “y” axes are not lisible. You should also add in the title the meaning of each light treatment (CK, FR-3 and FR-6).
Answers: Thank you. The figure quality has been improved and the information of lighting treatments has been added. Please see Figure 6 in the main text Line 551-556.
2) Fig. S2: you should enhance the quality the legend is not clear.
Answers: Thank you. The figure quality has been improved. Please see it in Figure S1.
3) Table S4: you should add the webpage link of each protein database.
Answers: Thank you. The webpage links have been added. Please see the revision in Table S3.
Many thanks for your comments and suggestions.

Reviewer 3 Report
The work submitted by Li et al. was well carried out in both experimental design and data analysis. Minor revision is needed for improvement.
1) Introduction needs to include any previous research progress in this specific plant species in the area of light quality treatments, if there is any.
2) Genome sequences are available in Brassica oleracea (wild cabbage), I wonder if your RNA-seq data can be mapped to the genome to provide another way of analyzing RNA-seq data.
3) I suggest to move the Supplementary Figure 1 and Table one to the main text, as these data provide more detailed and clearer information for treatments.
4) The term of "plant display" is used a few times, I do not this term, cannot find its use in related scientific literature, please explain.
5) Line 17: "different expressed genes" - "differential" is more often used. Line 132 - "SLA" - please define what it is.
6) Line 154 -"by by"
7) The term unigene is used for assembled transcripts, however, due to alternative splicing, using the transcript number to represent unigene number may not be accurate, blast cluster may give you an accurate estimate.
8) Fig 2(d), what the number (1 - 10000) means in the outmost circle?
9) Line 243 "fa-red"?
10) Figure 3, what the values mean? fold-change?
11) Line 353 "as energy to survivor," - needs correction.
12) Line 520-521: it is not a complete sentence, need to revise.
13) In discussion, just curious are there any genes of plant hormone receptors differentialy expressed? The leaf angle is controlled in maize by BR (brassinosteriod hormones), a recent paper published in Science, may have a look.
14) To promote re-use the data and increase citations of the publication, I like to know if the RNA-seq raw data are deposited into a public database, such as NCBI SRA database? The assembled transcripts are also suggested to be deposited into a public database, such as the following: https://www.ncbi.nlm.nih.gov/genbank/tsa/
Author Response
Replies to the Reviewer3’s comments
Thank you for your recognition on our research. This manuscript has been revised carefully according to your comments and suggestions. All the revisions were clearly highlighted in red. The replies to your comments are listed as follows:
- Introduction needs to include any previous research progress in this specific plant species in the area of light quality treatments, if there is any.
Answers: Thank you for this kind suggestion. Our group have several publications regarding to the effects of light quality on Chinese kale. However, most of our previous research focused on the red, blue, and UV-A light. Only one work involved in FR-induced growth of Chinese kale baby leaves was cited according to your suggestion. Please see it in Line 68-69.
- Genome sequences are available in Brassica oleracea (wild cabbage), I wonder if your RNA-seq data can be mapped to the genome to provide another way of analyzing RNA-seq data.
Answers: Thank you for pointing this question. It’s true that genome of Brassica oleracea may be a good choice when we study subspecies or varieties of wild cabbage. We can provide the raw data of RNA-seq in NCBI so that people who are interested in other ways of mapping.
In fact, we have unpublished genome of Chinese kale. However, the assembly quality of the unpublished genome seems to be improvement. Therefore, we choose to use de novo referring to Wu et al. (2017, doi: 10.3389/fpls.2017.00092) who got ideal sequencing result in Chinese kale.
- I suggest to move the Supplementary Figure 1 and Table 1 to the main text, as these data provide more detailed and clearer information for treatments.
Answers: Thank you for making our manuscript better. We have moved Supplementary Figure 1 and Table 1 to the main text. Please see them in Figure 6 (Line 552-556) and Table 4 (Line 557-561).
- The term of ‘plant display’ is used a few times, I do not this term, cannot find its use in related scientific literature, please explain.
Answers: Thank you for pointing this question. The ‘plant display’ means the maximum distance between blades under overhead view. We have added this definition in Section 4.3 (Line 566).
- Line 17: ‘different expressed genes’ - ‘differential’ is more often used.
Answers: Thank you very much. We have corrected the term. Please see the revision in Line 18.
- Line 132: ‘SLA’ - please define what it is.
Answers: Thank you. We have added this definition in Line 119.
- Line 154: ‘by by’.
Answers: Thank you. We have deleted it. Please see it in Line 146.
- The term unigene is used for assembled transcripts, however, due to alternative splicing, using the transcript number to represent unigene number may not be accurate, blast cluster may give you an accurate estimate.
Answers: Thank you. In Additional files: Table S2, the numbers of transcript (171879) and unigene (145746) have been listed. Among 145746 unigenes, 91.34% of unigenes can be annotated (Table S3). Please see the information in Additional files: Table S2-3.
- Fig 2(d): what the number (1 - 10000) means in the outmost circle?
Answers: The outmost circle could be regarded as an uneven Y-axis. For instance, the plant hormone signal transduction in Fig. 2d has 2340 genes in total and 38 DEGs in our result, so the light orange bar exceeds the Y-axis number 1000 and the deep purple bar exceeds the Y-axis number 10 while does not exceed number 100.
- Line 243: ‘fa-red’?
Answers: Thank you. We have corrected this spelling. Please see it in Line 245.
- Figure 3: what the values mean? fold-change?
Answers: Yes, the values indicate fold-change and we have added the explanation in Figure 3. Please see it in Line 304-306.
- Line 353: ‘as energy to survivor’ - needs correction.
Answers: Thank you. It has been corrected. Please see it in Line 361.
- Line 520-521: it is not a complete sentence, need to revise.
Answers: Thank you. This sentence has been revised. Please see it in Line 531-533.
- In discussion, just curious are there any genes of plant hormone receptors differentially expressed? The leaf angle is controlled in maize by BR (brassinosteriod hormones), a recent paper published in Science, may have a look.
Answers: Thank you for this question. The expressed biosynthesis genes and signal transduction genes were all listed in Figure 3c-d. We also have considered the BR-induced leaf tropism, however, our RNA-seq result showed that the biosynthesis genes of BR did not express in Chinese kale. Therefore, we did not focus on BR contents or its related pathways.
- To promote re-use the data and increase citations of the publication, I like to know if the RNA-seq raw data are deposited into a public database, such as NCBI SRA database? The assembled transcripts are also suggested to be deposited into a public database, such as the following: https://www.ncbi.nlm.nih.gov/genbank/tsa/.
Answers: We are grateful for this kind suggestion. We have uploaded all raw data according to your suggestion. And the SRA accession number is currently being checked by NCBI, which may take several days. Once the accession number is confirmed and released, we will promptly add it to the manuscript before publication.
Many thanks for your comments and suggestions.

Reviewer 4 Report
This manuscript describes the effect of far red light (3 W/m2 and 6 W/m2) on the physiological parameters of Chinese kale, demonstrating a significant increase compared to control plants. This was accompanied by improved photosynthetic parameters and increased content of some minerals. Next, the authors attempted to relate the findings to the processes taking place at molecular level, using various methodological approaches from RNA sequencing to analysis of the composition and content of 16 phytohormones. As a result, 1409 different expressed genes from 17 metabolic pathways were identified. And forms of hormones whose accumulation increased in plants under far red light were identified. The authors conclude that the additional presence of far red light can be a useful tool to regulate vegetative architecture, increase growing density, enhance photosynthesis, accumulate minerals, accelerate growth and produce significantly higher yields of Chinese cabbage.
Despite the many results presented, I feel that questions remain regarding the purpose of this study. The aims and objectives of this study are not well explained, given the biotechnological focus of the study, according to the author's conclusions. Especially for biotechnological tasks where the results can be applied. Please give details in introduction about the use in biotechnology of similar research. If the authors conclude (line 21) that “The results manifested that the supplementary far-red light can be a useful tool to regulate the vegetative architecture, elevate the density of cultivation, enhance the photosynthesis, benefit minerals accumulation, accelerate the growth, and obtain significant higher yield of Chinese kale”, so it should be comprehensive discussed in respect to current work and in relevance to previous studies. The summary of the results of the study in the conclusion of the article could have been better and more precise, giving more specific conclusions.
Author Response
Replies to the Reviewer4’s comments
Thank you for your time and efforts on our manuscript. This manuscript has been revised carefully according to your comments and suggestions. All the revisions were clearly highlighted in red. The replies to your comments are listed as follows:
1) The aims and objectives of this study are not well explained, given the biotechnological focus of the study, according to the author's conclusions. Especially for biotechnological tasks where the results can be applied. Please give details in introduction about the use in biotechnology of similar research.
Answers: Thank you for this suggestion. The biotechnology used in this study such as LEDs, morphological assay, photosynthetic traits measurement, and RNA-seq and bioinformatic analysis are commonly used in plant science. And similar works involve in far-red light and Chinese kale were rare.
In our study, the results such as narrowed leaf angle, increased biomass, and shortened flowering time by far-red light supplementation help to cultivate in high density, obtain higher yield per unit land, and to harvest earlier. These provided referential significance and could be applied quickly and directly in vegetable production.
From our view, the responses of these traits or parameters to far-red light have been explained in detail in Introduction section. In order to make our manuscript more concise, we have revised the Introduction section.
2) If the authors conclude (line 21) that “The results manifested that the supplementary far-red light can be a useful tool to regulate the vegetative architecture, elevate the density of cultivation, enhance the photosynthesis, benefit minerals accumulation, accelerate the growth, and obtain significant higher yield of Chinese kale”, so it should be comprehensive discussed in respect to current work and in relevance to previous studies.
Answers: Thank you for making our manuscript more concise. We have added the new citation. Please see it in Line 440-443.
3) The summary of the results of the study in the conclusion of the article could have been better and more precise, giving more specific conclusions.
Answers: Thank you for making our manuscript more precise. We have added the specific summary of value of the study. Please see the revision in Conclusion section (Line 683-686).
Many thanks for your comments and suggestions.

Round 2
Reviewer 4 Report
The authors have responded to all comments sufficiently. One suggestion is to exclude 0 W-m-2 (line 10) in the abstract, because the subsequent conclusion ( line 11:...were largely elevated..) is inconsistent with the light not supplemented by the far-red light.